# The Effect of Family Fertility Support Policies on Fertility, Their Contribution, and Policy Pathways to Fertility Improvement in OECD Countries

**DOI:** 10.3390/ijerph20064790

**Published:** 2023-03-08

**Authors:** Ting-Ting Zhang, Xiu-Yun Cai, Xiao-Hui Shi, Wei Zhu, Shao-Nan Shan

**Affiliations:** 1School of Public Finance and Taxation, Capital University of Economics and Business, Beijing 100070, China; 2School of Economics and Management, Shanxi Normal University, Taiyuan 030092, China; 3Institute of Industrial and Economic Policy, Beijing Economic and Technological Development Zone (BDA), Beijing 100070, China; 4School of Business, Shenyang University, Shenyang 110064, China

**Keywords:** family welfare policy, fertility rate, regression analysis, GRA, fsQCA

## Abstract

The cost of childbirth has been confirmed as a vital factor in families’ fertility decision-making, and family welfare policies are capable of compensating for the increase in household living expenses regarding childbirth, such that the country’s fertility situation can be optimized. In this study, the fertility promotion effects of family welfare policies in OECD(Organization for Economic Co-operation and Development) countries are investigated through regression analysis, grey correlation (GRA), and the fuzzy set qualitative comparative analysis fsQCA method. As indicated by the results: (1) Family welfare policies notably boost fertility, and the boosting effect is long-lasting. However, this boost will be weakened in countries where fertility rates remain below 1.5. (2) The contribution of welfare policy measures to the fertility-promotion effect varies by country. The contribution of cash benefits is highest in over half of the countries worldwide, the contribution of relevant services and in-kind expenditure is highest in 29% of the countries, and that of tax incentive expenditure is highest in 14% of the countries. (3) The policy mix to boost fertility also varies according to the social context, with three policy groups derived using the fsQCA method. To be specific, the core antecedent conditions comprise cash benefits, relevant services, and in-kind expenditure. On that basis, China should pay attention to the following three points when formulating family welfare policies to tackle their demographic challenges. First, a system of family welfare policies should be developed as early as possible in the context of increasingly severe demographic issues since the incentive effect of family welfare policies will be weakened in countries with chronically low fertility rates. Second, the effects of improvements vary by country, and China should comprehensively consider its national circumstances when formulating and dynamically adjusting the mix of government fertility support policies in accordance with its social development. Third, employment is the main means of securing family income and takes on critical significance to sustaining families. Unemployment exerts a significant disincentive effect, such that it is imperative to reduce youth unemployment and enhance the quality of youth employment. On that basis, the disincentive effect of unemployment on fertility can be reduced.

## 1. Introduction

As industrialization and urbanization have progressively deepened, the demand for labor tends to be reduced, and the cost of childbirth has risen, thus causing lower fertility rates. As a result, population issues have turned out to be a common issue in numerous countries. The global fertility rate will drop to 2.2 in 2050, and population growth will generally be decelerated in the decade ahead, as estimated by the United Nations World Population Outlook. Global population issues are becoming increasingly exacerbated. The OECD average fertility rates were already below the replacement level of 2.1 in 1984. The average fertility rate declined from 2.06 to 1.65 between 1984 and 2003, and it has rebounded since 2003 to roughly 1.69. However, the average fertility rate has been constantly below the population replacement level. As of 2019, fertility rates in 21 countries of the organization were lower than the international population threshold of 1.5 (Figure 1). To be specific, the fertility rate of Germany was less than 1.5 in 1975, such that Germany has been recognized as the first of the OECD countries to achieve a fertility rate below the international alert line. The second is Luxembourg, which achieved a fertility rate of 1.48 in 1976. From 1980 to 1999, 18 countries achieved fertility rates below 1.5, as well as Finland in 2017 and Norway in 2020. Currently, the industrialization level of China has lagged behind that of developed countries, whereas the number of births has been falling sharply since 2017. The data originating from China’s seventh census suggested that China’s total fertility rate fell to 1.3 in 2020, significantly below the international warning line. As revealed by the above-described result, a considerable number of countries are subjected to the dilemma of a long-term declining population (Van De Kaa, DJ, 1987 [1]). A sustained low fertility rate will result in a reduced working-age population, impose increased pressure on social security, and cause a lack of innovation in society, thus impeding economic growth and triggering other issues. Lastly, economic and social development cannot be sustained. Accordingly, raising the fertility rate has become an urgent problem to be solved in many countries.

OECD countries have formulated and implemented a series of welfare policies to stimulate fertility, so as to cope with the long-term population decline. A complete family welfare policy system has been formed based on long-term practice (Figure 2). Family benefit policies fall into three major types. The first type is cash benefits for family support, which comprise child allowances, benefits during parental leave, and benefits for single-parent families. The second type refers to relevant service subsidies (e.g., child care and early education facilities, child care services, and public spending on family services). The third is related tax incentives, which cover tax exemptions, child tax exemptions, and child tax credits.

However, the family welfare policies developed and implemented in OECD countries have different policy priorities and vary in their effectiveness. As depicted in Figure 3, the fertility rates in Germany, Italy, Japan and Korea were all lower than 1.5 in 2000. The fertility rates in all three countries fluctuated upwards and downwards from 2000 to 2019. After 20 years of development and policy adjustments, Germany’s fertility rate has reached over 1.5 in 2019, whereas Japan and Italy’s fertility rates remain below the alert level. Notably, Korea’s fertility rate is even below 1. In contrast, the fertility rates in the United States have been above the alert level, whereas the above-described rates have tended to decline over the past few years. In France, the fertility rates have remained largely stable. As revealed by the above-mentioned analysis, different countries have different policy priorities, and the effects of their policies vary significantly.

China, subjected to a declining birth rate, has begun to take measures to actively tackle the problem of low fertility. There are two main aspects of the measures: one is to lift birth restrictions, and the other is to provide tax incentives. The policy stipulating that couples who are both only children can have two children was rolled out in China in November 2011 to liberalize birth control. The implementation of the policy of allowing a couple to have two children if one of them is the only child in their family, i.e., the “separate two-child” policy, was launched on 15 November 2013, as clearly stated by the “Decision of the Central Committee of the Communist Party of China on Several Major Issues of Comprehensively Deepening Reform”. In 2015, the fifth plenary session of the 18th Central Committee decided to fully implement the policy that a couple can have two children, which is recognized as the aim to fully liberalize the “two-child policy”. Moreover, a couple of childbearing age can have three children, as stipulated by the “Decision on Optimizing the Fertility Policy for Long-term Balanced Population Development” of the political bureau meeting of the CPC Central Committee in 2021. In terms of tax benefits, the standing committee of the 13th National People’s Congress adopted a decision on amending the Individual Income Tax law in 2018, in which a deduction of CNY 1000 per person per month was stipulated for children’s education. Effective 1 January 2022, expenses regarding the care of taxpayers under the age of three would be deducted at a flat rate of CNY 1000 per month per taxpayer, as provided by the “Notice of the State Council on the Establishment of Special Additional Deductions for Personal Income Taxes for the Care of Infants and Children Under the Age of 3” on 28 March 2022.

The implementation of the above-described policies has had some effect (Figure 4). After the change in the birth limitation policy in 2011, the number of births in 2012 increased by 310,000 compared with 2011. After the implementation of the “separate two-child policy”, the number of births in 2014 increased by 470,000 compared with 2013. The number of births in 2016 increased by 1.31 million compared with 2015 after the full liberalization of the “two-child policy”. Nevertheless, China’s births fell off a cliff starting in 2017, down 630,000 from 2016, and in 2018, down 2 million from 2017, with another record-low birth rate in 2019, down 580,000 from 2018. This policy effect is short-term in nature, and the long-term effect is slight. Among women in China of childbearing age, 77.4% do not plan to have more children due to the heavy financial burden, as indicated by the results of the National Health Planning Commission’s 2017 sample survey on the national fertility status. The tax policies introduced in 2019 to reduce the economic burden of childbirth have not curbed the decline in births. This is a decline of 2.65 million in 2020 and 1.38 million in 2021. Notably, this policy has failed to address the root of the low fertility problem. Thus, policies that reduce the cost of childbearing for families should be urgently formulated to effectively address this challenge.

In brief, despite the similar welfare policies of OECD countries in response to fertility challenges, their policy effects vary considerably. Accordingly, when formulating future fertility support policies in China, the following issues should be considered.

(1)Whether family welfare policies have a boosting effect on fertility and whether the effect is long-term.(2)From a holistic perspective, this study analyzes how the combination of fertility support policies in different countries works better under different economic development conditions.(3)The formulation of relevant fertility support policies considering China’s actual situation.

Based on the above issues, the effect of family welfare policies on fertility is studied by regression analyses; the contribution of each policy to fertility is investigated through grey correlation, and then the optimal policy combination is found through a fuzzy set qualitative comparative analysis from a holistic perspective. In light of China’s actual situation, this provides experience and inspiration for the formulation and systematic construction of China’s welfare policy.

The rest of this study is organized as follows. In Section 2, the literature is reviewed. In Section 3, a theoretical framework and hypotheses are presented. In Section 4, the methods adopted in this study are introduced (e.g., regression analysis, grey relation analysis, and the fsQCA method). In Section 5, the results and discussions are presented. Lastly, in Section 6, the conclusions of this study are drawn.

## 2. Literature Review

There has been a wealth of research on the effects of family maternity support policies. The established literature can be divided into two categories based on the subject of the articles: the effect of family welfare policies on fertility rates; and how the effect of family welfare policies varies across countries.

### 2.1. The Effect of Family Welfare Policies on Fertility

The cost of childbearing is an important factor in household decision-making, and the increase in household expenditure resulting from childbearing increases with the age of the children, with food and education expenditure accounting for the largest share of additional costs (Emmanuel EA, Francis KA, Naa AS, (2022) [2]). Galindev, R (2011) [3] suggests that an increase in the cost of child rearing relative to the cost of leisure goods in the parental utility function leads to a decrease in fertility. Andersen, S.N.; Drange, N.; Lappegård, T (2018) [4] point out that the effect of family policy on fertility behaviour depends on the income effect of the policy and the opportunity cost of childcare to parents. Family policy focuses on the family or family member as a social unit and attempts to provide practical guidance in relation to social welfare (Kojima, H (1985) [5]). As a result, governments have developed fertility support policies to compensate for the increased cost of living for families, such that their fertility decision-making is affected. For instance, Hussey, LS (2010) [6] notes that extensive family leave laws are associated with lower abortion rates. Bae, Gwang, and Kim (2012) [7] show that government fertility policies have a catalytic effect on increasing fertility. High welfare benefits are capable of notably increasing the birth rate (Matthews, S; Ribar, D; Wilhelm, M (1997); Chai, G.-M [8,9]). Conversely, lowering welfare increments during the period of maternity benefits can reduce fertility rates (Horvath-Rose, AE; Peters, HE; Sabia, JJ (2008) [10]). In addition, the childcare support system has a significant positive effect on fertility (Ryu Yeongyu (2005) [11]). increasing spending on child welfare and care, as well as increasing the employment rate of women who have already had children, has been found to potentially elevate the level of fertility in Korea (Choi, Sang Joon; Myungsuk, Lee, (2013) [12]). In contrast, countries (e.g., France and Norway) that have maintained high levels of fertility have formulated active and comprehensive family support policies and employment security for women (Ellingsaeter, AL; Pedersen, E (2013); Toulemon, L; Pailhe, A; Rossier, C (2008) [13,14]). Some scholars have examined the key factors contributing to high fertility rates to clarify the social responsibility for upbringing, creating a social environment that reconciles family and work, providing childcare services through public services and satisfying the need for the country to bear the necessary costs of childbirth (Kim Seon-nyeo (2016) [15]), so as to achieve the non-exclusion and equal distribution of public resources (Yan, W (2017) [16]). Yunkyu and Ryoo (2014) [17] also suggested that improving childcare services can lower the risk of low fertility rates.

### 2.2. The Effect of Family Welfare Policies Varies across Countries

Jung, Kim, and Lim (2019) [18] analyzed the effects of family welfare policies on fertility in OECD countries through their research on countries in different regions. They highlighted that the effects of family welfare policies vary by region. To be specific, the fertility-promoting effects of family allowances are the most pronounced in northern Europe, whereas the effects of parental leave are the most pronounced in southern/eastern Europe. Notably, the effect of parental services is significant to a certain extent in eastern Europe and Asian countries. Yun (2015) [19], on the other hand, indicated a strong correlation between childcare arrangements and fertility. Based on the analysis of data on early childcare arrangements through a cluster analysis, childcare arrangements in 17 countries are classified into five categories (i.e., public de-familization, public de-familization, public and private de-familization, private de-familization, and private de-familization). Public de-familization achieved the highest fertility rates and the minimum class differences, while private de-familization achieved the lowest fertility rates and the maximum class differences. At different stages of development, welfare policies should be adjusted dynamically. Fanti, L; Gori, L (2010) [20] analysis shows that developed countries with below-replacement fertility and stagnant incomes can increase per capita income and fertility by increasing public spending on education rather than relying on child allowances. For instance, the traditional cash subsidy method in Germany is not effective at this stage, and a more effective solution to the fertility problem in Germany at this stage is to strengthen work–family coordination (Lee, Jinsook; Kim, Taewon (2014) [21]). Subsequently, Germany has witnessed a rebound in fertility rates by improving labor laws and the social security system in a family-friendly manner, expanding childcare facilities and supporting work and childcare (Kim, Young-Ran (2018) [22]). Kim, Kwangwoong (2012) [23] identifies the high cost of raising children and education; a social environment that does not allow women to combine work and family; and the precariousness of female employment as the causes of Korea’s low fertility. The main measures taken by Korea to cope with low fertility are the provision of childcare and enhanced maternity protection (Seo, J (2019) [24]). Through the practice of multi-country policy shows that cash subsidies have a greater incentive effect than time subsidies, and that a combination of cash and time subsidies is the most effective; the more timely the intervention, the more effective the fertility support policy; and the more intense the fertility support policy, the more effective the policy (Chen, M.; Zhang, m.; Shi, Z (2021) [25]). Rovny, AE (2011) [26] found that active labour market policies, generous work and family policies encourage higher fertility rates. Thus, a number of factors need to be taken into account to increase fertility (Aboulghasem, P.; Ahmad, S.; Mostafa, A.-R.; Rahim, K.-Z.; Jafari, H (2021) [27]).

In brief, an analysis of the existing literature reveals that family welfare policies can promote fertility, and the effect of family welfare policies on fertility varies considerably by country. Moreover, the analysis of the effects of family welfare policies on fertility is generally conducted using traditional econometric methods or through qualitative analysis. A traditional econometric analysis follows the independence of variables and the symmetry of cause and effect, thus ignoring the interrelationships between policies and failing to indicate the asymmetry of cause and effect. Furthermore, only the effect of individual welfare policies on fertility is analyzed, instead of the effect of policy groups on fertility. In contrast, a qualitative analysis is capable of analyzing the effects of individual policies and policy groups from a holistic perspective, whereas it has limited data handling for time series and panel data, such that analytical limitations are generated. Table 1 presents a summary and analysis of the above literature.

In this study, quantitative and qualitative research are integrated to deepen the analysis of the long-term and heterogeneous nature of the effect of family welfare policies on fertility through the analysis of the effect of family welfare policies on fertility. Moreover, the contribution of individual national welfare policy programs is measured before categorizing countries through a fuzzy set qualitative comparative analysis to identify more effective family-benefit policy combinations in different contexts.

## 3. Theoretical Background and Hypotheses

### 3.1. A Revised Theoretical Framework for New Household Economics

The “New Family Economics”, which emerged in the 1950s, is an economic method for analyzing the costs and benefits of household decisions, with Harvey Leibenstein and Gary Stanley Becker as its main representatives. The theory takes the family as the object of study while providing an economic analysis of fertility, marriage, and other behaviors to maximize the realization of utility by allocating the limited resources of the family in a rational manner. When analyzing family reproductive decisions, the theory considers children as a family’s durable goods and analyzes changes in the family’s demand for children through the analysis of the family’s demand for durable goods (Becker (2004) [29]. Maximizing household utility signifies how to allocate the available income wisely to purchase consumer durables and raise children under the determined household income, so as to maximize utility. Thus, the marginal cost of childbearing is a vital factor in a family’s decision to have children, which is elucidated as follows: first, the direct cost of clothing, food, lodging, and childcare expenses necessary for the growth of children; second, the indirect cost of childbearing, which is the loss of income from work and the loss of human capital due to parents leaving the workplace for childbearing and childcare (Gary S. Becker, 2004; James R. Walker, 1994) [29,30]. Andersen, Drange, and Lappegard (2018) elucidated the new family economics by asserting that changing the cost of childbearing has a certain effect on people’s fertility decision-making. Family welfare policies are capable of regulating the cost of childbearing in different manners; cash benefits, relevant service benefits, and tax incentives can lead to a reduction of the shadow price of childbearing (James R. Walker, 1994) [30]. As revealed by their findings, family welfare support policies are capable of compensating for the up-regulated costs of childbearing, whereas different policy measures can lead to reduced costs regarding raising children in different ways, such that the fertility rate can be increased. The fertility promotion effects of family welfare policies and which policy combinations provide better incentives are investigated in this study in accordance with the new family economics theory. Figure 5 presents a revised framework diagram for the new economics of the family, i.e., the theoretical framework applied in this study. Figure 5 illustrates how the new family economics compares children to consumer durables and how decision-making is based on the comparison between the cost of consumer durables and the cost of raising children. The government can increase the willingness of families to have children by increasing family welfare spending, offering cash support to families and relevant services, and in-kind expenditure and tax breaks to lower the cost of raising children, such that the total fertility rate can be elevated.

### 3.2. Research Hypothesis

As revealed by the above theory and the established literature, family welfare policies help to improve a country’s fertility situation, such that hypothesis 1 is formulated: public expenditure on family welfare has an elevating effect on fertility. Nevertheless, the implementation of family welfare policies differs from country to country in terms of the focus of government public expenditure, which is dependent on the specific measures covered in the policy, such that the resulting enhancement effect varies according to the country. Hypotheses 2 and 3 are thus formulated. Hypothesis 2: Each policy contributes differently to improving fertility. Hypothesis 3: The combination of family welfare policy measures to boost fertility varies across realities. The relevant variables in the text are shown in Table 2.

## 4. Methods and Data

### 4.1. Research Method

#### 4.1.1. Fixed Effects Regression Model

In this study, the effect of family welfare policies on fertility enhancement is analyzed through empirical regression. Regressions have been extensively used to analyze the effects of various factors on fertility in accordance with the established literature such as Choi and Myungsuk (2013) [12] and Jung, Kim, and Lim (2019) [18], all of whom have employed regressions to analyze the boosting or suppressing effects of various indicators on fertility. The assumption (i.e., that individual unobservable and non-time-varying variables are correlated with other variables due to fixed effects) is a more realistic assumption than that of random effects models and combined cross-sectional models. Based on this study, a regression model of family welfare policies and fertility is constructed, which is written as:(1)$$Y_{it}=\alpha_{i}+\beta P_{it}+\lambda X_{it}+T_{t}+\theta_{i}+\varepsilon_{it}$$
where i denotes the country; *t* is the year; $Y_{it}$ represents the fertility rate of a country in the t year; $P_{it}$ expresses the core explanatory variable, which represents the public expenditure on family benefits; β is a measure of the effect of family welfare policy on fertility, suggesting the net effect of family welfare policy after excluding the effect of time trends and individual differences; $X_{it}$ denotes the control variables (e.g., the unemployment rate, the mean age of women at childbirth, maternity leave, paternity leave, urbanization, the child-rearing ratio, medical expenditure, and the GDP per capita); $\alpha_{i}$ represents an unobservable random variable; $T_{t}$ represents time-fixed effects; $\theta_{i}$ represents individual fixed effects; and $\varepsilon_{it}$ is the model perturbation term, which contains other effects on fertility.

#### 4.1.2. Grey Relational Analysis (GRA)

Relevance analysis has been commonly used for data analysis in the grey system theory. The grey correlation analysis refers to a research method that describes the magnitude and order of the relationship between different indicators. If the correlation is higher, the factor more significantly affects the explanatory variable, and vice versa. The method requires fewer sample data and does not require the number of sample data, the pattern of data distribution, or the existence of a linear relationship. Thus, this method is adopted to measure the contribution of family welfare policy measures to fertility increases. The effects of cash welfare expenditure, relevant services and in-kind expenditure, and tax incentives on changes in fertility are examined using a grey correlation model. This reveals which individual country measure contributes most strongly to the fertility-raising effect. The analysis sequence is determined prior to the grey correlation calculation. The dependent variable sequence is expressed as the reference sequence $Y_{i}\left(t_{k}\right)=\left\{Y_{i}\left(t_{1}\right),Y_{i}\left(t_{2}\right),\cdots ,Y_{i}\left(t_{n}\right),\;k=1,2,\cdots n\right\}\;$(i=1,2,⋯n). Besides, the sequence of independent variables refers to the comparison sequence $X_{ij}\left(t_{k}\right)=\left\{X_{ij}\left(t_{1}\right),X_{ij}\left(t_{2}\right),\cdots ,X_{ij}\left(t_{3}\right),\;k=1,2,\cdots n\right\}\;$(i=1,2,⋯n, j=1,2,⋯n). The calculation steps are elucidated as follows. First, the data is standardized, and a comparative series of the reference variable dependent variable and the welfare policy measures is created. Second, the difference series is calculated and the absolute value of the difference between the reference series and the comparison series is determined. Third, the grey correlation coefficient is calculated. Lastly, the grey correlation degree is determined.
(2)$$\varepsilon_{ij}\left(t_{k}\right)=\frac{minmin\left|Y_{i}\left(t_{k}\right)-X_{ij}\left(t_{k}\right)\right|+\rho maxmax\left|Y_{i}\left(t_{k}\right)-X_{ij}\left(t_{k}\right)\right|}{\left|Y_{i}\left(t_{k}\right)-X_{ij}\left(t_{k}\right)\right|+\rho maxmax\left|Y_{i}\left(t_{k}\right)-X_{ij}\left(t_{k}\right)\right|}$$
(3)$$\gamma_{ij}=\frac{\sum_{k=1}^{n}\varepsilon_{ij}\left(t_{k}\right)}{n}$$i denotes the research subject and indicates the number of countries in the research sample; j represents the number of specific family welfare policy items. In this study, family welfare policies comprise three items. $t_{k}$ expresses time. $\varepsilon_{ij}\left(t_{k}\right)$ denotes the correlation coefficient between the family welfare policy item j and the total fertility in country i. $\gamma_{ij}$ represents the grey correlation. ρ is the discrimination coefficient, which is generally set to 0.5.

#### 4.1.3. Fuzzy Set Qualitative Comparative Analysis (fsQCA)

A qualitative comparative analysis is conducted through the analysis of set theory and Boolean algebra, which investigates how the combination of antecedent conditions results in the occurrence or appearance of the outcome variable from the perspective of the set. This analysis method refers to an integrated research method that combines the advantages of qualitative and quantitative research (Ragin, [31]). A qualitative comparative analysis is capable of analyzing the necessary and sufficient conditions for the outcome variable from a holistic perspective, and of generalizing the different combinations of antecedents that lead to the outcome variable, allowing the complexity of the antecedent conditions to be clarified. A traditional regression analysis, however, is susceptible to autocorrelation and multicollinearity and cannot simultaneously analyze the practical effects of multiple combinations of specific family welfare policy programs. As a result, a qualitative comparative analysis is highly advantageous in analyzing which policy combinations are more effective in raising fertility in different country contexts. On that basis, which combinations of family support policies promote higher fertility are analyzed through a qualitative comparative analysis.

QCA methods can fall into deterministic sets, fuzzy sets, and multi-valued sets, which is dependent on the form of the set. The fundamental difference between the above-mentioned three types is revealed in the different values taken for the division of the calibration affiliation of the variables. Conditional variables and outcome variables should be dichotomous variables, i.e., only 0 (no membership) or 1 (full membership), as required by the deterministic set qualitative analysis (CSQCA). A fuzzy set qualitative analysis (fsQCA) extends the deterministic set qualitative analysis to be able to cover affiliations in the middle of 0 and 1. Besides, a multi-valued set qualitative analysis (mvQCA) is extended from a deterministic set qualitative analysis between deterministic and fuzzy set states, which adds information to the variables by multi-segmenting their values based on the deterministic set. Thus, the qualitative analysis of fuzzy sets is more consistent with the reality of the situation. For instance, Bong, Jeon, and Seung (2021), Lin, and Kim (2016), and Yunkyu (2014) [17,28,32] qualitatively analyzed the effects of combinations of antecedent variables on fertility using fuzzy sets. Furthermore, a fuzzy set qualitative comparative analysis method is adopted in this research on the best combination of family welfare policies to enhance fertility based on the existing literature and the advantages of fuzzy set qualitative analysis.

### 4.2. Data Sources

#### 4.2.1. The Result Variable Total Fertility Rate (TFR)

The total fertility rate of 21 OECD countries from 2001 to 2015 serves as the result variable in this study. Figure 6 presents the total fertility rates of selected OECD countries in 2001 and 2015. In the figure, 21 countries are classified as low-fertility countries in accordance with the number of years in which the total fertility rate has been below the international fertility alert line (total fertility rate below 1.5) for over 13 years. The opposite is true for high-fertility countries. As depicted in Figure 6, most countries achieved higher total fertility rates in 2015 compared to 2001, with the Czech Republic, Germany, and Slovenia among the low-fertility countries rebounding to exceed the 1.5 level. Seven countries achieved a lower total fertility rate in 2001, of which Luxembourg among the high-fertility countries achieved a fertility rate reduced to 1.47. The above-described data originates from the OECD Family Database.

#### 4.2.2. Core Explanatory Variables

The core explanatory variable was public expenditure on family benefits. Figure 7 presents the average value of public expenditure on family welfare and the share of the respective item in GDP between 2001 and 2015. Public expenditure on family benefits comprises three types (i.e., cash benefits expenditure, family relevant services and in-kind expenditure, tax incentive expenditure). France achieves the highest total public expenditure on family benefits, accounting for 3.59% of GDP, followed by Luxembourg at 3.57%; South Korea has the lowest total public expenditure on family benefits. Germany has the highest expenditure on tax incentives, France achieves the highest expenditure on family services and in-kind expenditure, and Luxembourg is characterized by the highest expenditure on cash benefits. As revealed by the above results, countries implement family welfare policy measures with different priorities for financial support, such that families are supported for childbirth in different ways.

#### 4.2.3. Control Variables

The control variables were the unemployment rate, the mean age of women at childbirth, maternity leave, paternity leave, urbanization, the child-rearing ratio, medical expenditure, the GDP per capita, women’s education, as well as household income.

The unemployment rate was measured using the unemployment rate of the population aged 15–64 in the selected OECD countries from 2001–2015. Figure 8 illustrates the average unemployment rate for the population aged 15–64 in the selected OECD countries from 2001–2015. As depicted in Figure 8, Korea has the lowest average unemployment rate of only 3.62%, while Spain has the highest average unemployment rate with an average of 15.95%. The above-described data originate from the OECD Employment Database.

For the mean age of women at childbirth, Figure 8 presents the average age at childbearing for females in the selected OECD countries between 2001 and 2015. Among the selected OECD countries, Ireland has the oldest female age at childbearing at 31.27, whereas the Slovak Republic achieves the youngest female age at childbearing with an average age at childbearing of 28.11. The above-mentioned data originate from the OECD Household Database.

Parental leave was measured using the number of weeks of maternity leave and paternity leave in the selected OECD countries from 2001 to 2015. Figure 9 presents the number of weeks of maternity leave and paternity leave in the selected OECD countries in 2015. As depicted in Figure 9, the number of weeks of maternity leave exceeds that of paternity leave in all countries. Estonia has the longest maternity leave, with 166 weeks of maternity leave, and the longest paternity leave is in Korea, at 52.6 weeks. Meanwhile, the USA has the shortest maternity leave and paternity leave, both at 0 weeks.

Urbanization is the share of the urban population in the total population of the selected OECD countries from 2001 to 2015. The average urbanization rate of the selected OECD countries from 2001 to 2015 is depicted in Figure 10. The average urbanization rate of the selected OECD countries is above 50%, with the highest urbanization rate being in Belgium, which has an average urbanization rate of 97.5%, followed by Japan and Luxembourg; the lowest urbanization rate is in Slovenia with an average urbanization rate of 52.3%.

The child-rearing ratio is the ratio of the population under 15 years to the working-age population in the selected OECD countries from 2001 to 2015. The average child dependency ratio in the selected OECD countries from 2001 to 2015 is depicted in Figure 10. The country with the highest child support ratio is the United States, with both the US and Ireland having child support ratios above 30%; the country with the lowest child support ratio is Slovenia.

Medical expenditure is an indicator of the total medical expenditure as a share of GDP in the selected OECD countries from 2001 to 2015. The average total expenditure on health care in the selected OECD countries from 2001 to 2015 is depicted in Figure 11. The highest health care expenditure was in the USA with an average of 15.4%, followed by France (10.7%); the lowest health care expenditure was in Estonia with an average of 5.6%.

Gross domestic product (GDP) per capita for the selected OECD countries from 2001 to 2015 is used to represent the level of economic development of the respective country. Figure 11 shows the average GDP per capita for the selected OECD countries, 2001–2015. Figure 11 shows that Luxembourg has the highest GDP per capita with a GDP per capita of USD 105,879.8 and Poland has the lowest GDP per capita with a GDP per capita of USD 17,087.82. This variable is in US dollars using 2015 as the base purchasing power parity. The data originated from the OECD database of national economy statistics.

Household income is defined as the income of a minimum wage worker for a couple without children whose partner is not working. The household income divided by the income of an otherwise identical family working at the average wage. Figure 12 presents the income of a minimum wage worker for a couple without children whose partner was not working in 2015. As depicted in the above figure, Luxembourg has the highest income for households without children and the US is the lowest. The data originate from the OECD database.

Women’s education or female education refers to the female gross enrolment rate in tertiary education. Figure 12 illustrates the gross enrolment rate in tertiary education in 2015. Australia achieves the highest gross enrolment rate and Luxembourg has the lowest. The above data originate from the OECD database.

## 5. Results and Discussion

In this chapter, the effects of family welfare policies on fertility, the long-term effects of family welfare policies on fertility, and the effects of heterogeneity are presented using a regression analysis. The correlation between specific programs of family welfare policies and fertility is analyzed using grey correlation. Subsequently, the fsQCA method is employed to analyze how specific programs of welfare policy can be combined in different contexts to help increase fertility.

### 5.1. Statistical Description of Variables

Table 3 lists the descriptive statistics for the variables. The data listed in the table represent the overall statistical indicators for the respective variable, which comprise the sample size, standard deviation, minimum value, and maximum value of the variables. The data employed in this study are for 21 OECD countries between 2001 and 2015.

### 5.2. Analysis of the Effects of Family Welfare Policies on Fertility

#### 5.2.1. Baseline Regression of the Effect of Family Welfare Policies on Fertility

Table 4 lists the results of the benchmark regression analysis of the effect of family welfare policies on fertility. To be specific, columns (1) and (2) list OLS regressions, and column (3) represents a fixed-effects regression of the panel data. The results in columns (1), (2), and (3) show that family welfare policies notably boost fertility. The results of the fixed-effects regression of the panel data after the inclusion of control variables in column (3) suggest that the coefficient of public expenditure on family benefits reaches 0.069, suggesting that every 1% increase in public expenditure on family benefits leads to an elevation of the fertility rate by 0.069 units. As revealed by the above result, the adoption of a family welfare policy to address the low fertility challenge can effectively mitigate the continued low fertility rate and contribute to the optimization of a country’s fertility situation.

Considering the endogeneity of the core explanatory variable household welfare policy, this study also employs the method of an instrumental variable regression with the home government expenditure as the instrumental variable. In this study, two-stage least squares (2SLS) were used to test the instrumental variables, and the Kleibergen–Paap rk LM statistic was 15.4312 (*p* < 0.01), suggesting that the instrumental scalar was not unidentifiable, i.e., the instrumental variables passed the unidentifiability test. The Cragg–Donald Wald F statistic of 18.2704 exceeds the Stock–Yogo weak instrumental variable threshold of 10% at 16.38 and passes the weak instrumental variable test. The results of the two-stage regression are shown in column (4) of Table 4. Public expenditure on family benefits still significantly raises fertility and is significant at the 1% level, so the hypothesis that family welfare policies have the power to raise fertility is tested.

#### 5.2.2. Testing the Long-Term Effects of Family Welfare Policies on Fertility

To verify whether family welfare policies have a long-term effect on fertility, a N-year moving average of the explanatory variable, fertility, and the core explanatory variable, public expenditure on family welfare, are adopted to smooth out the short-term fluctuation components. The above data are regressed on panel data fixed effects, and the regression results are listed in Table 5. The results of the fixed effects regressions in Table 5 are the results of fixed effects regressions after a 3-, 4-, 5- and 6-year moving average treatment of the explanatory variable, fertility, and the core explanatory variable, public expenditure on family welfare. As depicted in Table 5, the regression coefficient for family welfare policies is still significantly positive and decreasing each year, suggesting that family welfare policies have a long-term persistent effect on fertility, not just a short-term effect. The regression coefficient of public expenditure on family welfare decreases each year, suggesting that the effect of family welfare expenditure on fertility exerts a long-term persistent effect, although the effect tends to be weakened over time.

#### 5.2.3. Heterogeneity Analysis

A heterogeneity analysis is conducted in this study for the total fertility situation across countries to analyze the effect of family welfare policies in different fertility scenarios. Countries can fall into the category of low-fertility countries and high-fertility countries in accordance with the number of years in which their total fertility rate has been below the international fertility alert line (total fertility rate below 1.5) in 2001–2015, with low-fertility countries being those with a total fertility rate lower than 1.5 for over 13 years; the remaining countries fall into the category of high-fertility countries. Table 6 lists the results of the heterogeneity analysis. Column (1) presents the results of the fixed-effects regression for high-fertility countries, while column (2) lists the results of the fixed-effects regression for low-fertility countries. As indicated by the results in columns (1) and (2), the regression coefficient of family welfare policies is 0.0748 at a significance level of 1% in high-fertility countries, whereas it is 0.0699 at a significance level of 5% in low-fertility countries. The above results suggest that the effects of family welfare policies vary according to fertility rates. Family welfare policies are more effective in high-fertility countries and less effective in low-fertility countries. China is a high-fertility country based on the above-mentioned classification, therefore China should improve its family welfare policy as soon as possible to curb the continuously reducing fertility rate.

### 5.3. Analysis of the Contribution of Family Support Policies to Fertility Rates and Policy Groups

Public expenditure on family benefits falls into three types, i.e., cash benefits, expenditure on goods and services, and expenditure on tax concessions. As indicated by the baseline regression analysis of 5.2, family welfare policies notably boost fertility. In this section, the effect of the three policy programs on fertility is further analyzed. The association between the three types of expenditure and the total fertility rate in the respective country, i.e., the contribution of the three types of expenditure to the improvement of the total fertility rate, is investigated using the grey correlation. Furthermore, how the combination of family welfare policy programs is more effective in boosting fertility in a variety of country contexts is analyzed using the fsQCA method.

#### 5.3.1. Grey Correlation between the Type of Family Benefits Expenditure and the Total Fertility Rate

Table 6 lists the correlation between the three types of household welfare expenditure and the total fertility rate for the respective country for 2001–2015 using a grey correlation analysis, i.e., the grey correlation between cash welfare expenditure, relevant services and in-kind expenditure, and tax benefits and the total fertility rate, as well as the contribution of the three types of expenditure to the total fertility rate. The higher the correlation, the higher the contribution of that type of expenditure to the total fertility rate will be, and the higher the expenditure in that area, the greater the increase in the total fertility rate. The correlation calculations in Table 7 suggest that cash welfare expenditure is most highly correlated with total fertility in 12 countries, followed by relevant services and in-kind expenditure in six countries, and tax benefits are least highly correlated with fertility status in three countries. Thus, cash benefits are the most effective in promoting fertility in most countries, followed by relevant services and in-kind payments, and tax benefits. However, given the specificities of their macro-environment and national circumstances, the contribution of cash benefits expenditure, relevant services and in-kind expenditure, and tax incentives to fertility varies considerably by country.

#### 5.3.2. A Combination of Family Welfare Policies to Boost Fertility

Table 6 lists the extent to which different programs of family welfare policy affect fertility in individual countries, although this is an analysis of individual countries that is not generalizable. Thus, the social context of the respective country should also be considered when drawing on and analyzing the above results. On that basis, the mix of family welfare policies appropriate to the national context is investigated using the fsQCA method in this section.

##### Variable Calibration

In the qualitative comparative analysis of fuzzy sets, the respective variable is considered a set with a different degree of affiliation. On that basis, the variables should be calibrated, such that they are transformed into a fuzzy affiliation score, i.e., between 0.0 and 1.0, prior to the group analysis. The qualitative comparative analysis of fuzzy sets requires three qualitative anchor points for variables in the calibration process (i.e., setting full affiliation, crossover, and full disaffiliation). The percentile employed in most existing research is adopted in this study to determine the qualitative anchor points. Moreover, the 5th, 50th, and 95th percentiles of the sample data are identified to be fully unaffiliated, crossover, and fully affiliated in accordance with the characteristics of the data distribution. Furthermore, the sample data variables are calibrated (Table 8). The variable indicators are generally set to 3–8 during the qualitative histological analysis of fuzzy sets (Ragin [31]). In conjunction with this study, five indicators are selected for the histological analysis. The variables have served as measures of household welfare policies adopted by OECD countries and indicators that measure the social context of the country. The family benefits policy measures comprise cash benefits expenditure, relevant services and in-kind expenditure, and tax concessional expenditure. The unemployment rate and the child-rearing ratio are selected as the indicators of the social context of the country. The above-described two indicators are employed as a measure of the social context of the country since they take on a certain significance to fertility in the regression analysis, whether in the OLS regression, the fixed effects regression, or the instrumental variables regression.

##### Necessity Test

The necessity test is a test of whether the respective condition variable is required for the outcome variable, i.e., the occurrence of the outcome should arise from that condition variable. The necessity of each condition variable is studied by fsQCA3.0, and the results are listed in Table 9. The consistency of all individual variables in the test results is less than 0.9, suggesting that individual variables cannot constitute the necessary conditions for the outcome variable. It is noteworthy that the effect and influence of individual conditional variables are not sufficient to lead to an increase in fertility, such that the different groups of conditional variables should be analyzed to identify policy combinations that have the capability of stabilizing and increasing fertility.

##### Sufficient Conditions: Policy Portfolio Options

The combination of policies that can boost fertility from the above-described family welfare policies is analyzed using the fsQCA method. In this study, the fuzzy set calibration is performed in the same way as studies in the established literature, where the values of the corresponding indicators in the 5th, 50th, and 95th percentiles serve as the calibration parameters when calibrating the indicators. Moreover, the built-in coefficients of the fsQCA software system are employed (i.e., a consistency threshold of 0.8 and a case threshold of 1) for setting the consistency and case thresholds. The standard analysis is conducted to obtain three conformational solutions that are conducive to improving the low fertility situation (i.e., the complex solution, the parsimonious solution, and the intermediate solution). Furthermore, the most representative intermediate solution is adopted, and Ragin and Fiss’ presentation of the QCA results are drawn on, as listed in Table 10.

The respective column in Table 9 represents one of the possible groups. The consistency of all three conditional groups exceeds 0.8 (Table 9), suggesting that the consistency condition is satisfied in all cases. In other words, all three groups are sufficient conditions for increasing fertility.

C1: Higher burden of child support. This grouping states that “fertility rate = cash benefits * services and in-kind support * child-rearing ratio” and suggests that countries with a high child-support burden can increase their fertility rates by providing cash benefits and services and in-kind support to families. In this group, unemployment and family tax incentives do not affect the level of fertility rates, i.e., neither the level of unemployment nor the strength of family tax incentives affects fertility rates, provided the basic conditions of this group are satisfied. The United Kingdom, Belgium, Australia, France, and Ireland are involved as the countries in this grouping scenario.

C2: Low unemployment and high child-support burden. This group states that “fertility rate = services and in-kind support * tax incentives * ~unemployment rate * child-rearing ratio”. This group suggests that high services and in-kind support and high tax incentives are sufficient conditions to raise fertility rates in countries with low unemployment. Cash welfare policies do not have a significant effect on fertility rates when the basic conditions of the histogram are met. The countries in this group of scenarios are the Netherlands and the United Kingdom.

C3: Low unemployment rate and low burden of child support. This group states that “fertility rates = cash benefits*~services and in-kind support*tax incentives*~unemployment*~ child-rearing ratio”, suggesting that high cash benefits, high tax incentives, and low services and in-kind support can contribute to higher fertility rates when unemployment is low and the burden of child support is low. This group only involves one country, which is the Czech Republic.

## 6. Conclusions and Policy Implications

In this study, the effect of family welfare policies on boosting fertility is analyzed based on a systematic analysis of the panel data originating from OECD countries between 2001 and 2015 using several methods (e.g., regression analysis, grey correlation, and fuzzy set qualitative comparative analysis). The results of this study are presented as follows:(1)Family welfare policies are capable of significantly enhancing fertility status (see Table 3 for results). Moreover, the long-term sustainability of the fertility promotion effect of family welfare policies is verified by moving-mean regression (Table 4). The heterogeneity analysis suggests that differences exist in the boosting effects of family welfare policies under different fertility statuses. To be specific, family welfare policies have higher boosting effects in high-fertility countries than in low-fertility countries (Table 5).(2)The correlation degree between three types of family welfare policy and the fertility rate is calculated using the method of grey correlation degree analysis. In other words, cash welfare expenditure, relevant services and in-kind expenditure, and tax incentives and the fertility rate. It was found that cash benefit expenditure made the largest contribution to fertility improvement in 12 of 21 countries, relevant services and in-kind expenditure made the largest contribution to fertility improvement in 6 of 21 countries, and tax concessions made the largest contribution to fertility improvement in 3 of 21 countries (Table 6).(3)Through the qualitative analysis of fuzzy sets, three policy combinations are found to improve fertility status. As depicted in Table 9, expenditure on cash benefits and relevant services and expenditure in kind are the core conditions. Cash welfare expenditure appears in C1 and C3, and relevant service and physical expenditure appear in C1 and C2. The policy mix to improve fertility also varies in different social contexts.

In brief, increasing government subsidies to families can be conducive to increasing the total fertility rate. A grey correlation analysis and fsQCA qualitative analysis can be employed in the practice of family benefits policy to indicate the types of policies that public expenditure on family benefits should orientate towards. As indicated by the results of a grey relation analysis and fsQCA qualitative analysis, cash allowance expenditure, family service expenditure, and in-kind expenditure significantly increase the fertility rate, and they serve as the core conditions in the grouping of fertility rate enhancement as well. Accordingly, the support of cash benefits to families and that of family-associated services should be emphasized more considerably in practice. Furthermore, the policy mix should be adjusted dynamically in accordance with the macro-environment to curb the decline of the fertility rate.

OECD countries had an earlier start to deal with low fertility, and their policy systems and response measures are more refined. Thus, the measures taken by OECD countries can bring great enlightenment to China to assist China in addressing the problem of low fertility, though there are some differences between China and OECD countries in terms of the macro-environment when China faces its population problem.

(1)In the face of increasingly serious population issues, a family welfare policy system should be established as early as possible to improve the fertility situation. Because of the heterogeneity and long-term nature of the uplifting effects of family welfare policies, China’s early formulation and improvement of family welfare policies can have a greater effect in encouraging fertility.(2)The development of a family welfare policy requires an increase in expenditure on cash benefits and relevant services and in-kind expenditure. As indicated by the results of Section 5, cash benefit expenditure most significantly boosts fertility, such that cash support to families should be increased. Fertility rates can be more significantly elevated by increasing quality childcare services, childcare and early education facilities, and youth assistance to reduce family-associated expenditures and the financial pressure on families to raise children.(3)According to the macro-environment, the amount of the three types of public expenditure on family welfare should be dynamically adjusted. From the results of Table 10, it can be seen that the best mix of policies to promote fertility varies with unemployment and child support burdens. Against the background of a low unemployment rate, we need to choose the type of combination of family welfare public expenditure according to the context of the child-support burden. High child-support burdens require increased spending on services and in-kind and tax incentives to increase fertility, while low child-support burdens require increased spending on cash benefits and tax incentives. Different social contexts, therefore, require a constant refocusing of policy spending to respond effectively to the decline in fertility.

Despite the certain practical significance of this study, there are still certain defects and limitations. First, the research subject of this study is OECD countries with certain specificity. Given the applicability of the results of this study, this study should be considered in the context of the national conditions of the respective country. Second, there are numerous factors involved in fertility decision-making. Lastly, the factors involved in family fertility decision-making change at different stages of development, such that the formulation and implementation of welfare policies should be dynamically adjusted. Accordingly, the above-described three aspects can be studied in depth in subsequent research.

## Figures and Tables

**Figure 1 ijerph-20-04790-f001:**
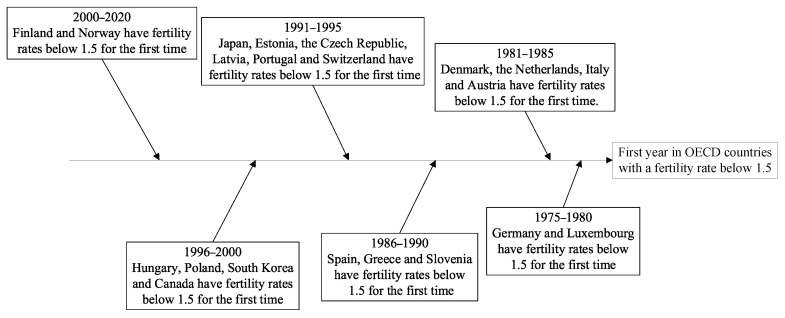
Countries in the OECD with fertility rates below 1.5 for the first time. Data source: OECD family database.

**Figure 2 ijerph-20-04790-f002:**
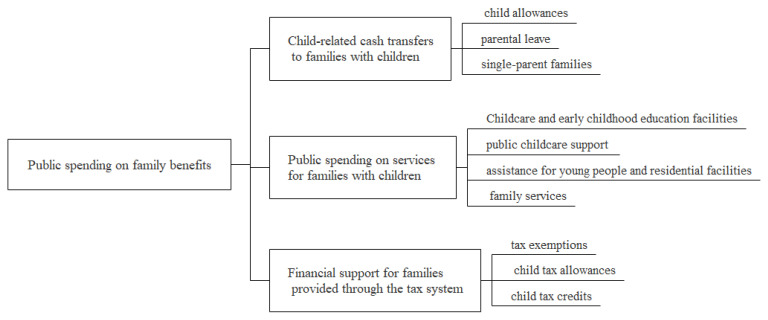
Family welfare policies in OECD countries.

**Figure 3 ijerph-20-04790-f003:**
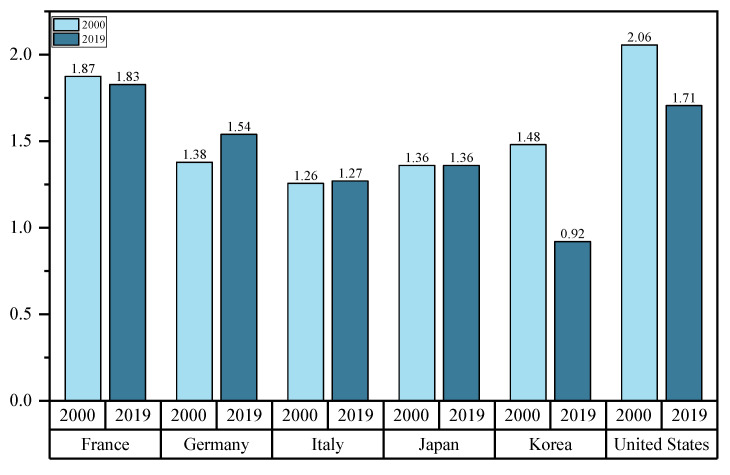
Fertility trends of different countries.

**Figure 4 ijerph-20-04790-f004:**
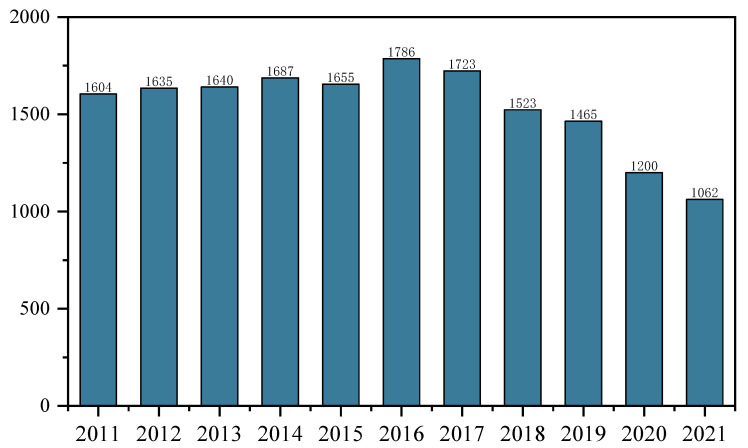
Number of births in China, 2011–2021 (10,000). Data source: China National Economic and Social Development Statistical Bulletin, 2011–2021.

**Figure 5 ijerph-20-04790-f005:**
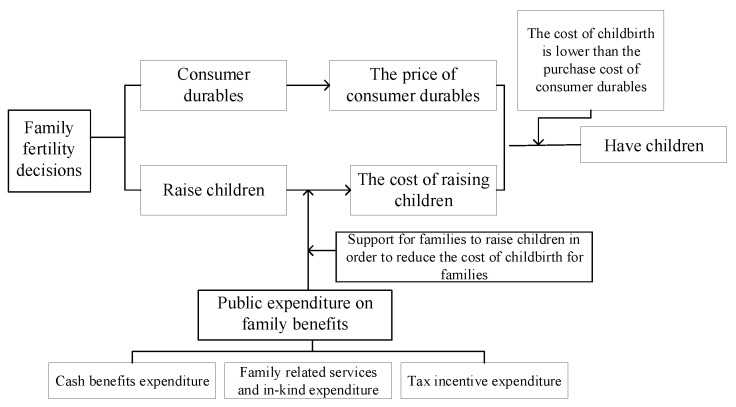
Modified “New Home Economics” framework diagram.

**Figure 6 ijerph-20-04790-f006:**
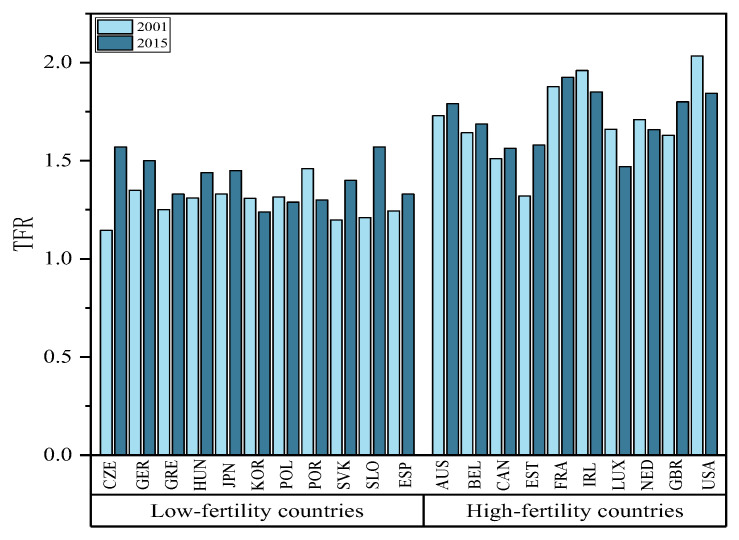
Total fertility rates in the selected OECD countries, 2001 and 2015.

**Figure 7 ijerph-20-04790-f007:**
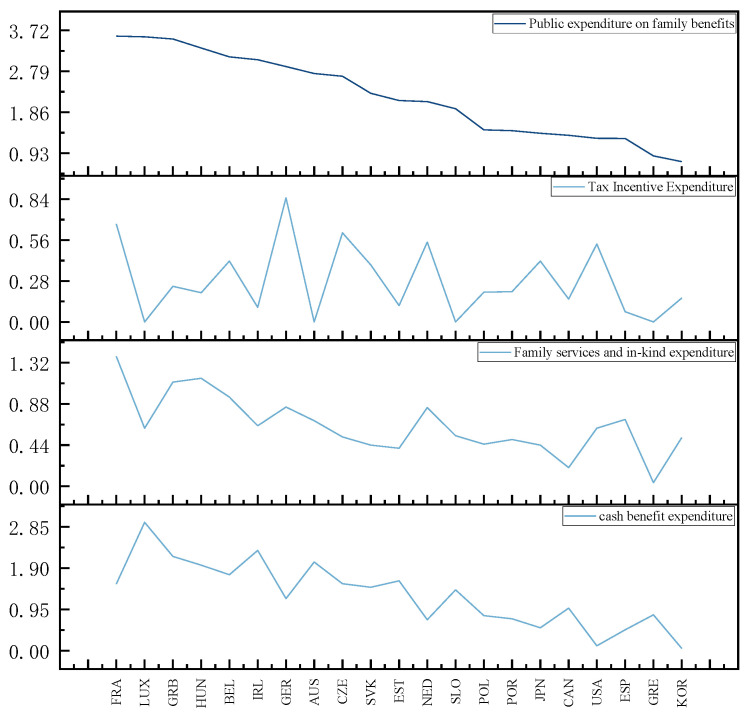
Average expenditure on total family welfare and average expenditure on three items in the selected OECD countries, 2001–2015.

**Figure 8 ijerph-20-04790-f008:**
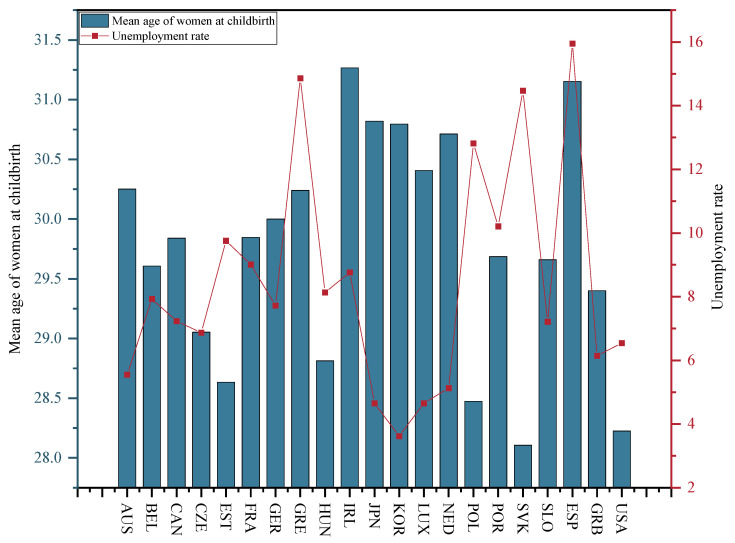
Average unemployment rate in the selected OECD countries, 2001–2015.

**Figure 9 ijerph-20-04790-f009:**
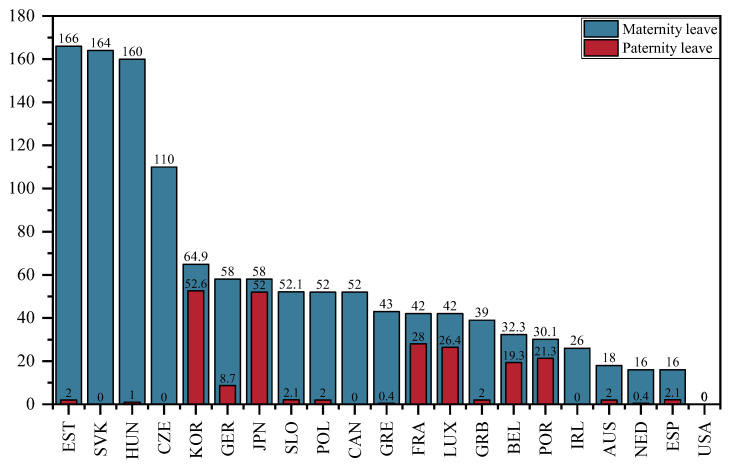
Number of weeks’ maternity leave and paternity leave of in the selected OECD countries, 2015.

**Figure 10 ijerph-20-04790-f010:**
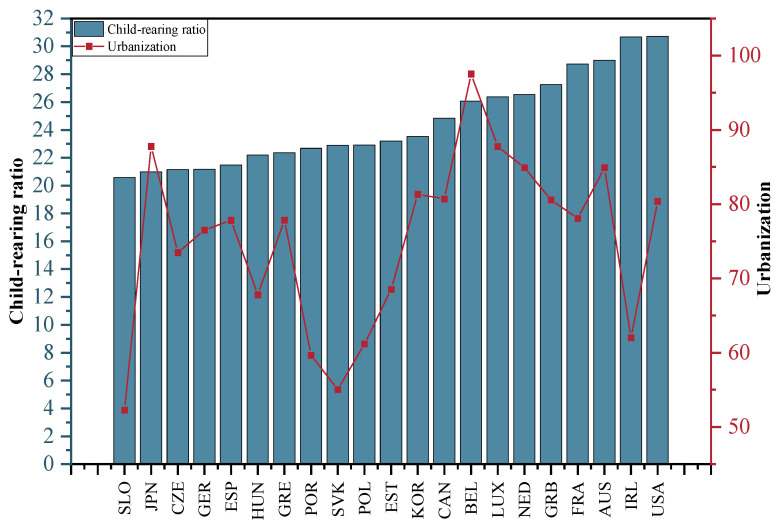
Average urbanization rate and child-rearing ratio in the selected OECD countries, 2001–2015.

**Figure 11 ijerph-20-04790-f011:**
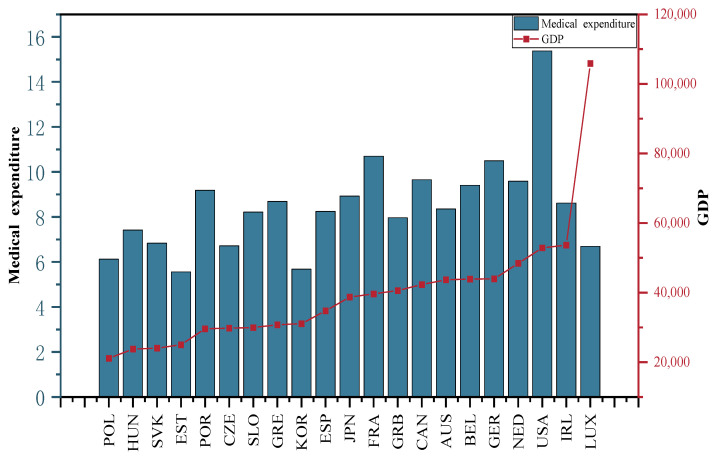
Average GDP per capita and medical expenditure in the selected OECD countries, 2001–2015.

**Figure 12 ijerph-20-04790-f012:**
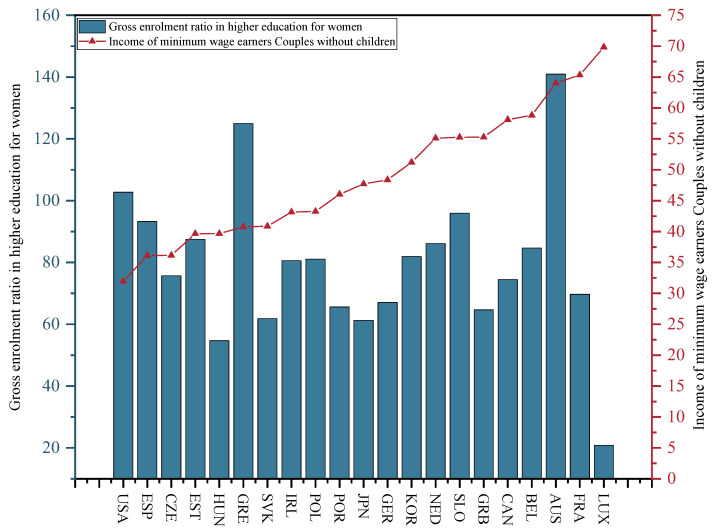
Gross enrolment ratio in higher education for women and the income of minimum wage-earning couples without children in the selected OECD countries in 2015.

**Table 1 ijerph-20-04790-t001:** A summary of the most relevant research.

Ref.	Year	Method	Shortcoming	Finding
Choi and Myungsuk [12]	2013	The effect of local policies on fertility in Korea was analyzed using a fixed effects regression model.	First, in the quantitative analysis, the above-mentioned measures, ignoring the interrelationships between policies, do not analyze the effects of policy groups on fertility. Second, in the qualitative analysis, there is limited handling of time series and panel data, which affects the accuracy of the analysis results.	First, in this study, a fuzzy set analysis is used to remedy the shortcomings in the traditional measurement methods by adding an analysis of the effect of policy groups on fertility. Second, this study uses a fixed-effects regression analysis to determine the relationship between individual welfare policies and fertility before conducting a policy group analysis to provide a basis for the policy group analysis.
Bae and Kim [7]	2012	Using a logit model hierarchical analysis, the study showed that government fertility policies can promote fertility.		
Yun [19]	2015	The relationship between the type of childcare system and inter-class fertility was analyzed using a cluster analysis.		
Jung, Kim and Lim [18]	2019	Using a dynamic panel analysis, the implementation effects of family policies to boost fertility in OECD countries were analyzed.		
Ellingsaeter and Pedersen [13]	2013	Interviews with workers revealed that their fertility intentions were closely related to their own economic conditions.		
Bong, Jeon and Seung [28]	2021	A fuzzy set comparative analysis was used to find the social characteristics of high-fertility countries		
Toulemon, Pailhe and Rossier [14]	2008	A comparative analysis illustrates the reasons for the high and stable fertility rate in France.		

**Table 2 ijerph-20-04790-t002:** Variable definitions and descriptions.

Type of Variable		Variable Name	Variable Descriptions
Result variable		Total fertility rate	The average number of children born per woman over a lifetime given the current age-specific fertility rates and assuming no female mortality during the reproductive years.
Explanatory variables	Core explanatory variables	Public expenditure on family benefits	Public spending on family benefits includes financial support that is exclusively for families and children. Public expenditure on household welfare as a percentage of GDP is used to denote this indicator.
	Control variables	Unemployment rate	The unemployment rate of the population aged 15–64 in the selected OECD countries from 2001–2015.
		Mean age of women at childbirth	The mean age of mothers at birth, calculated as the simple mean average age in years of women at childbirth.
		Maternity leave	The length of paid maternity, parental, and home care leave available to mothers in weeks.
		Paternity leave	The length of paid paternity and parental leave reserved for fathers in weeks.
		Urbanization	The share of the urban population in the total population.
		Child-rearing ratio	The share of the population under 15 years of age in the working age population.
		Medical expenditure	Total medical expenditure as a share of GDP.
		GDP per capita	Gross domestic product per capita.
		Women’s education	The female gross enrolment rate in tertiary education.
		Household income	The income of a minimum wage worker for a couple without children whose partner is not working.

**Table 3 ijerph-20-04790-t003:** Descriptive statistics of variables.

Variable	Unit of Measurement	Mean	Std. Dev.	Min	Max	Observations
Total fertility rate (TFR)	Number of individuals	1.548046	0.261449	1.085	2.12	315
Public expenditure on family benefits	Percentage (%)	2.22454	0.95964	0.2	4.3	315
Unemployment rate	Percentage (%)	8.439729	4.590629	1.8091	27.6954	315
Mean age of women at childbirth	Mean age	29.76157	1.097209	26.8	32.23	315
Maternity leave	Weeks	56.89619	51.75062	0	166	315
Paternity leave	Weeks	6.390476	12.57735	0	52.6	315
Medical expenditure	Percentage (%)	8.497483	2.259661	4.395	16.816	315
Urbanization	Percentage (%)	0.750538	0.11683	0.508	0.97877	315
Child-rearing ratio	Percentage (%)	24.53664	3.412558	18.787	33.4771	315
GDP per capita		39,692.18	17,737.37	15,911.68	114,804.6	315
Household income	Percentage (%)	59.78393	13.03358	33.39	85.93	315
Women’s education	Percentage (%)	71.17668	20.66297	10.4083	140.9518	315

**Table 4 ijerph-20-04790-t004:** Baseline regressions of family welfare policies to increase fertility.

Variable	(1) OLS	(2) OLS	(3) FE	(4) IV
Public expenditure on family benefits	0.1250992 ***	0.10235568 ***	0.07279541 ***	0.3040572 ***
Unemployment rate		−0.0094394 ***	−0.008107 ***	−0.01373959 ***
Mean age of women at childbirth		0.0136427 *	−0.0050052	−0.09784289 **
Maternity leave		−0.0000939	−0.00104557 **	−0.00081465
Paternity leave		−0.001637 ***	0.00052812	0.0006837
GDP per capita		0.1106857 ***	0.04908031	−0.04906729
Medical expenditure		0.0218199 ***	−0.00490288	−0.03737607 **
Urbanization		−0.0247392	−0.93256182 ***	−0.40280507
Child-rearing ratio		0.04184921 ***	0.02321994 ***	0.02399765 ***
Household income		−0.0018602 ***	−0.00017588	0.000198
Women’s education		0.0031895 ***	0.00054916	0.00223738 **
National fixed	NO	NO	Yes	Yes
Year fixed	NO	NO	Yes	Yes

Legend: * *p* < 0.1; ** *p* < 0.05; *** *p* < 0.01.

**Table 5 ijerph-20-04790-t005:** Tests of the long-term effects of family welfare policies (moving mean regression).

Variable	3-Year	4-Year	5-Year	6-Year
Public expenditure on family benefits	0.08068017 ***	0.07937807 ***	0.07110307 ***	0.06583601 ***
Unemployment rate	−0.0053786 ***	−0.00399635 **	−0.00297958	−0.00236795
Mean age of women at childbirth	−0.03627623	−0.05496354 **	−0.07805347 ***	−0.10766642 ***
Maternity leave	−0.001032 **	−0.00111621 **	−0.00107736 **	−0.00095538 **
Paternity leave	0.00053154	0.0005187	0.00069277	0.00080207 **
GDP per capita	0.00570146	−0.02199649	−0.1131876	−0.22537668 **
Medical expenditure	−0.00317853	−0.00380275	−0.00641123	−0.00824315
Urbanization	−1.1706674 **	−1.2805105 **	−1.7348818 ***	−2.1990592 ***
Child-rearing ratio	0.0186022 ***	0.01623977 ***	0.01457493 ***	0.01283534 ***
Household income	0.00049051	0.00107996 *	0.00131975 **	0.00158262 ***
Women’s education	0.00051576	0.00033213	−0.00009076	−0.00046868
National fixed	Yes	Yes	Yes	Yes
Year fixed	Yes	Yes	Yes	Yes

Legend: * *p* < 0.1; ** *p* < 0.05; *** *p* < 0.01.

**Table 6 ijerph-20-04790-t006:** Heterogeneity analysis test.

Variable	(1) High-Fertility Countries	(2) Low-Fertility Countries
Public expenditure on family benefits	0.07801336 ***	0.05372713 *
Unemployment rate	−0.00591429	−0.00857392 ***
Mean age of women at childbirth	−0.04380867	0.02588732
Maternity leave	−0.00040761	−0.00152903 **
Paternity leave	0.00160217 *	−0.00088584 *
GDP per capita	0.12534204	0.16627189
Medical expenditure	−0.0253916	0.01906103 **
Urbanization	−0.84669597	0.00190662
Child-rearing ratio	0.02472164 ***	0.02120663 **
Household income	−0.00427994 ***	0.00076725
Women’s education	−0.00091925	0.00179824 **
National fixed	Yes	Yes
Year fixed	Yes	Yes

Legend: * *p* < 0.1; ** *p* < 0.05; *** *p* < 0.01.

**Table 7 ijerph-20-04790-t007:** Grey correlation between three types of public expenditure on family benefits and the total fertility rate in OECD countries.

Type	Country	Cash Benefits Expenditure	Family Services and In-Kind Expenditure	Tax Incentive Expenditure
Highest correlation with cash expenditure	United Kingdom	0.9018	0.8463	0.5814
	Netherlands	0.8973	0.8171	0.5438
	Estonia	0.8678	0.7949	0.7840
	Spain	0.8641	0.8306	0.8257
	Slovak Republic	0.8490	0.7594	0.6061
	Poland	0.8440	0.6900	0.5156
	Czech Republic	0.8162	0.7894	0.6438
	Belgium	0.7906	0.5659	0.7179
	Luxembourg	0.7421	0.7318	——
	Germany	0.7378	0.6951	0.7212
	France	0.6881	0.5535	0.5166
	Greece	0.6278	0.6238	——
Highest correlation between services and in-kind expenditure	Canada	0.7523	0.9082	0.6050
	Japan	0.6423	0.9085	0.7811
	Australia	0.5766	0.7158	——
	Hungary	0.5522	0.7755	0.6910
	Slovenia	0.5467	0.5642	——
	United States	0.4501	0.7550	0.5880
Highest relevance of tax benefits	Ireland	0.7119	0.5586	0.9592
	Portugal	0.6776	0.6726	0.8643
	Korea	0.5874	0.5865	0.9560

Note: “—” represents the absence of this type of public expenditure in this country.

**Table 8 ijerph-20-04790-t008:** Qualitative anchor points for the respective variable.

Variable	Fully Affiliated	Crossover	Fully Unaffiliated
Cash benefits expenditure	2.306667	1.4	0.113333
Relevant services and in-kind expenditure	1.153333	0.62	0.2
Tax incentive expenditure	0.67	0.203333	0
Unemployment rate	14.8611	7.72432	4.650824
Child-rearing ratio	30.66589	23.20359	20.98739
TFR	1.97	1.419333	1.294933

**Table 9 ijerph-20-04790-t009:** Analysis of the necessity of conditional variables.

	Consistency	Coverage
cash benefits expenditure	0.749282	0.779880
relevant services and in-kind expenditure	0.719617	0.727273
tax incentive expenditure	0.597129	0.595420
unemployment rate	0.495694	0.530194
child-rearing ratio	0.850718	0.853167
~cash benefits expenditure	0.543541	0.518248
~relevant services and in-kind expenditure	0.528230	0.517824
~tax incentive expenditure	0.594258	0.590304
~unemployment rate	0.759809	0.707035
~child-rearing ratio	0.377033	0.372401

**Table 10 ijerph-20-04790-t010:** Grouping of family welfare policies to boost fertility.

	C1	C2	C3
Cash benefits expenditure	⚫		⚫
Relevant services and in-kind expenditure	⚫	⚫	⊛
Tax incentive expenditure		●	●
Unemployment rate		Ⓧ	Ⓧ
Child-rearing ratio	⚫	⚫	⊛
Consistency	0.917431	0.877551	0.832031
Coverage	0.478469	0.370335	0.203828
Unique coverage	0.213397	0.105263	0.0708134
Overall consistency	0.858218		
Overall coverage	0.654546		

Note: “⚫” and “●” represent the presence of a condition. “Ⓧ” and “⊛” suggest that the condition does not appear. “⚫” and “Ⓧ” denote core conditions. “•” and “⊛” denote marginal conditions. The core condition refers to the condition variables that appear in both the “simplex solution” and the “intermediate solution” of the QCA analysis and have a greater impact on the results. Marginal conditions are condition variables that appear only in the intermediate solutions of the QCA analysis. Marginal conditions are less important to the results compared with core conditions.

## Data Availability

The data used to support the findings of this study are available from the corresponding author upon request.

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
