# Peer review of "The Effect of Family Fertility Support Policies on Fertility, Their Contribution, and Policy Pathways to Fertility Improvement in OECD Countries"

_ijerph, 2023, doi:10.3390/ijerph20064790_

Round 1

Reviewer 1 Report

See attached PDF for suggestions. First suggestion is to re-title the paper slightly. In lines 387-388, you identify "total fertility rate" as the EXPLANATORY variable.

But in Figure 7, you show "total fertility rate" as the RESULTING variable. This is confusing/erroneous. 

The research model (Figure 7) suggests structural equation modeling. But the regression model suggests FAMILY WELFARE POLICY as a major explanatory variable. But, it appears that family welfare policy is operationalized as cash benefit expenditure, family services and in-kind expenditure, and tax incentive expenditures. These variables have slightly different nomenclature throughout the manuscript; thus, it is difficult to determine if the authors are changing nomenclature or changing VARIABLES. Names of variables should be FIXED. Any difference in subsequent use would indicate a DIFFERENT VARIABLE.

In lines 358-360, you write that "traditional regression analysis ... is based on the assumption that the variables are independent of each other..." That is simply not correct.

Figure 9 is unclear. On the left are words, "family," "tax," service," and "cash." How are these related to your important explanatory variables? 

LIne 389, you write "... rates of OECD countries" which would connote all of them. You should modify that with the adjective "selected." Otherwise, you've left out 15 or more OECD countries.

Throughout, then, you might consider modifying your "OECD countries" by inserting "low fertility" as a modifier.

In line 411-412, you write "... the share of each item in GDP for grandmothers..." Is that accurate? Nonsensical.

When you write (line 425) that "control variables include unemployment rate," I believe you intend to continue on with other control variables. The verb "include" connotes that there are ADDITIONAL control variables. But you should list them all. So, proper verb would be "are."

Figure 13 should have a "starting point" of 0 on the Y axis. Otherwise, the visual is misleading. Slovenia child-rearing ratio appears to be HALF of  of, say Korea, whn it's really only 21 versus 23! 

Figure 14 needs to be anchored with 0 as well. (Estonia is NOT less than half of Spain in medical expenditures, but that's what the bar chart implies.)

Table 2 and Table 3, first variable: "Family benefits." Is that the same as the "overall" family welfare policy/expenditures? In lines 517, 518 refer to "public expenditure on family benefits" and "public expenditure on household welfare." Are these three the SAME VARIABLE? If so, ARE THEY A COMBINATION of the 3 indicator variables from the first model?

Line 529, should not report p=.00. It's never 0. Better to report p<.001, for example.

Line 534, should that be Table 3 (rather than Table 2)? 

Lines 544 and 547, Table 4? (rather than Table 3).

In Table 2, line 315, "TFR" is used for the first time, I believe. It should be spelled out first with (TFR) in parentheses.

In line 574, this recommendation would be true if China is a "high fertility country" as those are the countries in which family benefits work best (according to lines 572-573 and Table 5.

The methods are unclear concerning the NUMBER of observations from which the correlations were calculated (Table 6). Table 2 suggests a total N of 315. That would average to 15 observations per country. But TFR in a country is exactly that--ONE NUMBER. So it's unclear how a correlation could be computed between 15 observations of cash payments in UK but ONE fertility rate. 

In Table 6, what two variables are correlated for each correlation coefficient? The title of the table reads "fertility status" (is that TFR?) and "family welfare policies." The left column reads "highest correlation WITH cash expenditure" and then the second one reads "highest correlation BETWEEN services and in-kind expenditure." The third entry reads "Highest relevance of tax benefits." UNCLEAR

In Table 9, there is a symbol O with * in it that is not explained. 

In lines 699-700, is it "countries" or a country (just one named--Czech Rep.).

In conclusions and implications, you've shown STATISTICAL significance, but what about PRACTICAL significance (EFFECT SIZE)? e.g., WHAT would it "take" to bring low fertility countries up to above 1.5? 

In line 725, you write "... were present i BOTH groupings (Table 9)." Were there not 3 groupings--C1, C2, and C3? 

Recommendations (1) and (3) (see lines 729-760 seem appropriate. (2) does not give much guidance. 

References need MUCH EDITING.

Finally, will you or will you not use the Oxford comma? Just be consistent.

Author Response

Dear Respected Reviewer,

Hello

Thank you very much for your attention. We also thank you for devoting your valuable time for reviewing our manuscript. The authors are sure that your comments improved the quality of the manuscript. We tried to answer all your comments carefully.

1)First suggestion is to re-title the paper slightly. In lines 387-388, you identify "total fertility rate" as the EXPLANATORY variable. But in Figure 7, you show "total fertility rate" as the RESULTING variable. This is confusing/erroneous.

Thanks to the reviewer for your valuable comments.

Based on your comments, we have made changes. "Total fertility rate" is the RESULTING variable in this paper. Figure 7 is therefore correct. We have changed the " EXPLANATORY variable " in lines 395-397 to the “RESULTING variable”.

2) The research model (Figure 7) suggests structural equation modeling. But the regression model suggests FAMILY WELFARE POLICY as a major explanatory variable. But, it appears that family welfare policy is operationalized as cash benefit expenditure, family services and in-kind expenditure, and tax incentive expenditures. These variables have slightly different nomenclature throughout the manuscript; thus, it is difficult to determine if the authors are changing nomenclature or changing VARIABLES. Names of variables should be FIXED. Any difference in subsequent use would indicate a DIFFERENT VARIABLE.

Thanks to the reviewer for your valuable comments.

Based on your suggestions, we have made certain modifications to Figure 6. FAMILY WELFARE POLICY, represented by public expenditure on family benefits, is the main explanatory variable in the regression model, which analyses the effect on total fertility. The three types of public expenditure on family benefits are cash benefits expenditure, related services and in-kind expenditure and tax incentive expenditure. The grey correlation analyses which type works better, while the fsQCA analyses how the three types can be combined in different social contexts to have a better contribution. Both methods are more in-depth analyses of the effects of public expenditure on family benefits.

3) In lines 358-360, you write that "traditional regression analysis ... is based on the assumption that the variables are independent of each other..." That is simply not correct.

Thanks to the reviewer for your valuable comments.

Based on your suggestion, we deleted this sentence and expressed the previous meaning in more detail. The details, as follows:” Traditional regression analysis, however, is susceptible to autocorrelation and multicollinearity and cannot simultaneously analyze the practical effects of multiple combinations of specific family welfare policy programs.”

4) Figure 9 is unclear. On the left are words, "family," "tax," service," and "cash." How are these related to your important explanatory variables?

Thanks to the reviewer for your valuable comments.

In Figure 8, Family refers to public expenditure on family benefits." tax," service," and "cash." refer to the three types of public expenditure on family benefits, which represent tax incentive expenditure, related services and in-kind expenditure and cash benefits expenditure respectively. The public expenditure on household benefits is the sum of these three types of expenditure. We have made changes to the icon in Figure 8.

5) Line 389, you write "... rates of OECD countries" which would connote all of them. You should modify that with the adjective "selected." Otherwise, you've left out 15 or more OECD countries. Throughout, then, you might consider modifying your "OECD countries" by inserting "low fertility" as a modifier.

Thanks to the reviewer for your valuable comments.

In response to your suggestion, we have amended the description in line 389 and also added the qualifier “selected “for OECD countries in the subsequent content.

6) In line 411-412, you write "... the share of each item in GDP for grandmothers..." Is that accurate? Nonsensical.

Thanks to the reviewer for your valuable comments.

As you say, the correct expression in this sentence is " Figure 8 shows the average value of public expenditure on family welfare and the share of each item in GDP, 2001-2015. ", having removed "for grandmothers" from the sentence.

7)When you write (line 425) that "control variables include unemployment rate," I believe you intend to continue on with other control variables. The verb "include" connotes that there are ADDITIONAL control variables. But you should list them all. So, proper verb would be "are."

Thanks to the reviewer for your valuable comments.

As you say, the control variables also include other control variables. We have listed all of the other control variables. Control variables include unemployment rate, Mean age of women at childbirth, maternity leave, paternity leave, urbanization, child-rearing ratio, medical expenditure, GDP per capita, women's education, and household income.

8)Figure 13 should have a "starting point" of 0 on the Y axis. Otherwise, the visual is misleading. Slovenia child-rearing ratio appears to be HALF of  of, say Korea, whn it's really only 21 versus 23!

Thanks to the reviewer for your valuable comments.

As per your suggestion, the starting point of the child-rearing ratio in Figure 11 has been changed to start from 0 in order to avoid misinterpretation.

Figure 11. Average urbanization rate and child-rearing ratio in selected OECD coun-tries, 2001-2015.

9)Figure 14 needs to be anchored with 0 as well. (Estonia is NOT less than half of Spain in medical expenditures, but that's what the bar chart implies.)

Thanks to the reviewer for your valuable comments.

As per your suggestion, the starting point of the medical expenditure in Figure 12 has been changed to start from 0 in order to avoid misinterpretation.

10)Table 2 and Table 3, first variable: "Family benefits." Is that the same as the "overall" family welfare policy/expenditures? In lines 517, 518 refer to "public expenditure on family benefits" and "public expenditure on household welfare." Are these three the SAME VARIABLE? If so, ARE THEY A COMBINATION of the 3 indicator variables from the first model?

Thanks to the reviewer for your valuable comments.

In response to your comments, Table2 and table3, the first variable has been modified and all of them have been changed to public expenditure on family benefits. In lines 517, 518 refer to "public expenditure on family benefits" and "public expenditure on household welfare". These names are the same variable and are thus unified as "public expenditure on family benefits", which is the sum of tax incentive expenditure, related services and in-kind expenditure and cash benefits expenditure.

11) Line 529 should not report p=.00. It's never 0. Better to report p<.001, for example.

Thanks to the reviewer for your valuable comments.

As per your suggestion, p=.00 has been replaced with p<.001 in the article. see line 539.

12) Line 534, should that be Table 3 (rather than Table 2)?

Thanks to the reviewer for your valuable comments.

Thank you for your corrections. Table3 is correct, we have now corrected table2 to table3. see line 544.

13) Lines 544 and 547, Table 4? (rather than Table 3).

Thanks to the reviewer for your valuable comments.

Thank you for your corrections. Table4 is correct; we have now corrected table3 to table4. see line 554 and 557.

14) In Table 2, line 315, "TFR" is used for the first time, I believe. It should be spelled out first with (TFR) in parentheses.

Thanks to the reviewer for your valuable comments.

Based on your suggestions," total fertility rate (TFR)" is spelled out in table2.

15) In line 574, this recommendation would be true if China is a "high fertility country" as those are the countries in which family benefits work best (according to lines 572-573 and Table 5.

Thanks to the reviewer for your valuable comments.

China belongs to the category of high-fertility countries according to the criteria for classifying heterogeneity. According to your advice,the type of China is thus described in lines 584-585. The details are as follows.

China is a high-fertility country according to the above-mentioned classification, so China should improve its family welfare policy as soon as possible to cope with the continued low fertility rate.

16) The methods are unclear concerning the NUMBER of observations from which the correlations were calculated (Table 6). Table 2 suggests a total N of 315. That would average to 15 observations per country. But TFR in a country is exactly that--ONE NUMBER. So it's unclear how a correlation could be computed between 15 observations of cash payments in UK but ONE fertility rate.

Thanks to the reviewer for your valuable comments.

In response to your comments, the formula for table6 has been corrected to describe the calculation process in more detail. The grey correlation is the average of the correlation coefficients of each factor with fertility over the years. The grey correlation between each factor and the fertility rate for 15 years is thus a single number. this is the research subject and indicates the number of countries in the research sample. is the number of specific family welfare policy items, and in this paper family welfare policies include three items.  indicates time.  is the correlation coefficient between family welfare policy item  and total fertility in country .  is the grey correlation and  is the discrimination coefficient, usually set to 0.5.

17) In Table 6, what two variables are correlated for each correlation coefficient? The title of the table reads "fertility status" (is that TFR?) and "family welfare policies." The left column reads "highest correlation WITH cash expenditure" and then the second one reads "highest correlation BETWEEN services and in-kind expenditure." The third entry reads "Highest relevance of tax benefits." UNCLEAR

Thanks to the reviewer for your valuable comments.

Based on your suggestion, in order to illustrate the contents of Table 6 more precisely, we have modified the table header toGrey correlation between three types of public expenditure on family benefits and total fertility rate in OECD countries.”. The table is presented according to the type of household welfare expenditure that makes the highest contribution to fertility, e.g., "highest correlation WITH cash expenditure" refers to countries where cash welfare expenditure makes the highest contribution to fertility. For example, the grey correlation between cash welfare expenditure and total fertility in the United Kingdom is 0.9018, the correlation between related services and in-kind expenditure is 0.8463, and the grey correlation between tax concessional expenditure is 0.5814, 0.9018>0.8463>0.5814, thus the United Kingdom is the country with the highest correlation with cash welfare expenditure.

18) In Table 9, there is a symbol O with * in it that is not explained.

Thanks to the reviewer for your valuable comments.

⊛" indicate that the condition does not appear. Based on your suggestions, an explanation of ⊛ has been added below table9.

19) In lines 699-700, is it "countries" or a country (just one named--Czech Rep.).

Thanks to the reviewer for your valuable comments.

Based on your suggestion, we have modified the sentence. Replace the original sentence with” Only one country, the Czech Republic, is in this grouping.”.

20) In conclusions and implications, you've shown STATISTICAL significance, but what about PRACTICAL significance (EFFECT SIZE)? e.g., WHAT would it "take" to bring low fertility countries up to above 1.5?

Thanks to the reviewer for your valuable comments.

Practical initiatives have been added to the conclusion, based on your suggestions. Including practical initiatives in the conclusion, based on your suggestions. In lines 746-757 of the article, the following practical initiatives have been added.

“In summary, it can be seen that increased government subsidies to fam-ilies can help improve the total fertility rate. In the practice of family benefits policy, grey correlation analysis and fsQCA qualitative analysis can be used to indicate the types of policies that public expenditure on family benefits should orientate towards. The results of grey relation analysis and fsQCA qualitative analysis show that cash allowance expenditure, family service expenditure and in-kind expenditure contribute to the fertility rate well, and they are also the core conditions in the grouping of fertility rate enhancement as well. Therefore, in practice, more emphasis should be placed on the sup-port of cash benefits to families and on the support of family-related services. At the same time, the policy mix should be adjusted dynamically according to the macro-environment to deal with the declining fertility rate.”

21) In line 725, you write "... were present i BOTH groupings (Table 9)." Were there not 3 groupings--C1, C2, and C3?

Thanks to the reviewer for your valuable comments.

In accordance with your suggestion, we have rephrased the paragraph to avoid misunderstanding. The paragraph was amended to” Through the qualitative analysis of fuzzy sets, three policy combinations are found to improve the fertility status. As can be seen from Table 9, expenditure on cash benefits and related services and expenditure in kind are the core conditions. The cash welfare expenditure appears in C1 and C3, and the related service and physical expenditure appear in C1 and C2. The policy mix to improve fertility also varies in different social contexts.”

22) Recommendations (1) and (3) (see lines 729-760 seem appropriate. (2) does not give much guidance.

Thanks to the reviewer for your valuable comments.

Based on your suggestions, we have modified recommendation 2 in order to make it more suitable. The modifications are as follows.

“The development of a family welfare policy requires an increase in expenditure on cash benefits and related services and in-kind expenditure. The results of Section5 show that cash benefit expenditure has the highest effect on fertility improvement, so it is necessary to increase cash support to families. As a result of the covid-19 pandemic in recent years, the unemployment rate has increased. More quality childcare services, childcare and early education facilities and youth assistance to reduce family-related expenditures and the financial pressure on families to raise children will have a better effect on raising fertility rates. Different social contexts require a constant refocusing of policy spending to respond effectively to the decline in fertility.”

23) References need MUCH EDITING. Finally, will you or will you not use the Oxford comma? Just be consistent.

Thanks to the reviewer for your valuable comments.

Based on your suggestions, we have retouched the references.

Reviewer 2 Report

1. The framework presented on page 9 should be briefly explained.

2. Equation 1 should be more explicit, and the constant parameter (alpha) should be explained along with others (page 10).

3. Figure 5 is not necessary in the manuscript. It occupies space unnecessarily.

4. Why is women's education, and household income not controlled in the analysis. These two variables are tantamount to fertility decline.

5. (Page 23), This is making a huge leap into the world of unknown. You cannot deduce conclusions based OECD countries for China. You may conduct a similar and compare results with that of OECD.  

6. Long sentences and minor typos need to be addressed.

7. Other comments are included in the attached manuscript.

Author Response

Dear Respected Reviewer,

Hello

Thank you very much for your attention. We also thank you for devoting your valuable time for reviewing our manuscript. The authors are sure that your comments improved the quality of the manuscript. We tried to answer all your comments carefully.

Reviewer#2:

1) The framework presented on page 9 should be briefly explained.

Thanks to the reviewer for your valuable comments.

Based on your suggestions, in lines 277-283, a description of Figure 5 has been added. The additions are as follows:

Figure 5 shows how the new family economics compares children to consumer durables and makes decisions by comparing the cost of consumer durables with the cost of raising children. By increasing family welfare spending, providing families with cash support, related services and in-kind and tax breaks to reduce the cost of raising children, the government increases the willingness of families to have children, thus increasing total fertility rate.

2) Equation 1 should be more explicit, and the constant parameter (alpha) should be explained along with others (page 10).

Thanks to the reviewer for your valuable comments.

Based on your suggestions, in lines 321-322, an explanation of the alpha is added.  is an unobservable random variable.

3) Figure 5 is not necessary in the manuscript. It occupies space unnecessarily.

Thanks to the reviewer for your valuable comments.

Based on your suggestions, we have removed Figure 5.

4) Why is women's education, and household income not controlled in the analysis. These two variables are tantamount to fertility decline.

Thanks to the reviewer for your valuable comments.

Based on your suggestions, variables representing women's education, and household income were added to the control variables. Women's education, which refers to the female gross enrolment rate in tertiary education. Household income, which is the income of a minimum wage worker for a couple without children whose partner is not working.

5) (Page 23), This is making a huge leap into the world of unknown. You cannot deduce conclusions-based OECD countries for China. You may conduct a similar and compare results with that of OECD.

Thanks to the reviewer for your valuable comments.

Based on your suggestions, in line 759-764, we have elaborated on the applicability of the family welfare policy to China. The details are as follows.

“OECD countries began to deal with low fertility earlier, policy systems and response measures are more perfect. Therefore, although there are some differences between China and OECD countries in the macro-environment when China faces the population problem, the measures taken by OECD countries can bring great enlightenment to China, to provide some assistance for China to deal with the problem of low fertility.”

6) Long sentences and minor typos need to be addressed.

Thanks to the reviewer for your valuable comments.

We have commissioned the team of professionals to touch up the language of this article and to correct all grammatical and spelling errors in the article to ensure the quality of the article.

Round 2

Reviewer 1 Report

Suggested edits:

Line 46: Remove "been"

Line 65: Remove "even"

Line 68: Remove "trends"

Lines 69 and 70: Change "the" to "a"

Line 79: Remove "currently"

Line 91: Remove "the"

Line 94: Should be punctuated as in 2019;

Line 105: Remove "down"

Lines 109 and 128: Remove "the"

Line 155: Change "from" to "From"

Line 164: Remove "the"

Line 177: After ";" insert "and"

Line 182: Replace "children" with "additional costs"

Figure 6 appears to be an explanatory/statistical model of variables/causes and effects. But closer examination suggests that it is more conceptual than the model of how variables will be analyzed. For instance, on the far left, you write "Explanatory variables" and then to the right you LIST what those explanatory variables are. I suggest that this be changed to a table or figure that depicts what the primary variables ARE--rather than how they will be analyzed in a path analytical framework.

Line 384: Replae "is" with "are"

Line 385: Remove "the"

Line 429: Replace "cover" with "were"

Line 523: Remove "the" after "using"

Table 2: What are the units of measurement? (e.g., Age at childbirth is easily seen as YEARS. But, what unit of measurement is "women's education" at 71.18?

Line 581: Fix legend (**p<.05  rather than **        p<.05)

Table 9: Identify what columns C1, C2, and C3 are.

Line 754: Rather than percentages, report numbers (e.g., rather than report "57% of countries," report "12 of 21 countries.") Do that for all 3.

Line 782: Replace "the" with "its"

Line 787: Replace "better" with "greater"

Line 797 on: This conclusion does not seem to be supported by findings directly from this study, especially the conclusion dealing with the quality of youth employment. If this is a conclusion or an IMPLICATION drawn from using the findings of OTHER research, cite that research.

Author Response

Dear Respected Reviewer,

Hello

Thank you very much for your attention. We also thank you for devoting your valuable time for reviewing our manuscript. The authors are sure that your comments improved the quality of the manuscript. We tried to answer all your comments carefully.

1) Line 46: Remove "been"

Thanks to the reviewer for your valuable comments.

Based on your comments, we have removed" been" from the sentence. Now this sentence is " and the cost of childbirth has risen,".

2) Line 65: Remove "even"

Thanks to the reviewer for your valuable comments.

Based on your comments, we have removed" even" from the sentence. Now this sentence is " total fertility rate fell to 1.3 in 2020,".

3) Line 68: Remove "trends"

Thanks to the reviewer for your valuable comments.

Based on your comments, we have removed" trends" from the sentence. Now this sentence is " a considerable number of countries are subjected to the dilemma of long-term declining population,".

4) Lines 69 and 70: Change "the" to "a"

Thanks to the reviewer for your valuable comments.

Based on your comments, we change the word "the" in the sentence to "a". Now this sentence is " A sustained low fertility rate will result in a reduced working-age population, impose an increased pressure on social security, cause a lack of innovation in society,".

5) Line 79: Remove "currently"

Thanks to the reviewer for your valuable comments.

Based on your comments, we have removed" currently" from the sentence. Now this sentence is " A complete family welfare policy system has been formed after long-term practice (Fig. 2). ".

6) Line 91: Remove "the"

Thanks to the reviewer for your valuable comments.

Based on your comments, we have removed" the" from the sentence. Now this sentence is " fertility rates in Germany, Italy and, Japan and Korea were all lower than 1.5 in 2000. ".

7) Line 94: Should be punctuated as in 2019;

Thanks to the reviewer for your valuable comments.

Based on your comments, we change the punctuated "," into";". Now this sentence is " Germany's fertility rate has reached over 1.5 in 2019; ".

8) Line 105: Remove "down"

Thanks to the reviewer for your valuable comments.

Based on your comments, we have removed" down" from the sentence. Now this sentence is " has begun to take measures to tackle the problem of low fertility actively. ".

9) Lines 109 and 128: Remove "the"

Thanks to the reviewer for your valuable comments.

Based on your comments, we have removed "the" from the sentence. Now the two sentences are " The implementation of policy of allowing a couple to have two children. , and "The implementation of above-described policies has had some effect (Fig. 4). ".

10) Line 155: Change "from" to "From"

Thanks to the reviewer for your valuable comments.

Based on your comments, we change the word " from " in the sentence to " From ". Now this sentence is " From a holistic perspective, ".

11)Line 164: Remove "the"

Thanks to the reviewer for your valuable comments.

Based on your comments, we have removed "the" from the sentence. Now this sentence is " In light of China's actual situation ".

12)Line 177: After ";" insert "and"

Thanks to the reviewer for your valuable comments.

Based on your comments, we have inserted "and" into the sentence. Now this sentence is " the effect of family welfare policies on fertility rates; and the effect of family welfare policies varies across countries.".

13)Line 182: Replace "children" with "additional costs"

Thanks to the reviewer for your valuable comments.

Based on your comments, we change the word " children " in the sentence to " additional costs". Now this sentence is "with food and education expenditures accounting for the largest share of additional costs".

14) Figure 6 appears to be an explanatory/statistical model of variables/causes and effects. But closer examination suggests that it is more conceptual than the model of how variables will be analyzed. For instance, on the far left, you write "Explanatory variables" and then to the right you LIST what those explanatory variables are. I suggest that this be changed to a table or figure that depicts what the primary variables ARE--rather than how they will be analyzed in a path analytical framework.

Thanks to the reviewer for your valuable comments.

Based on your comments, we change the content of Figure 6 to be presented as a table. 

15) Line 384: Replace "is" with "are"

Thanks to the reviewer for your valuable comments.

Based on your comments, we change the word " is " in the sentence to " are ". Now this sentence is "higher fertility are analyzed through qualitative comparative analysis. ".

16) Line 385: Remove "the"

Thanks to the reviewer for your valuable comments.

Based on your comments, we have removed "the" from the sentence. Now this sentence is " higher fertility are analyzed through qualitative comparative analysis.".

17) Line 429: Replace "cover" with "were"

Thanks to the reviewer for your valuable comments.

Based on your comments, we change the word " cover " in the sentence to " were ". Now this sentence is " Control variables were unemployment rate, Mean age of women at childbirth, maternity leave, paternity leave, urbanization, child-rearing ratio, medical expenditure, GDP per capita, women's education, as well as household income.".

18) Line 523: Remove "the" after "using"

Thanks to the reviewer for your valuable comments.

Based on your comments, we have removed "the" from the sentence. Now this sentence is " The correlation between specific programs of the family welfare policy and fertility is analyzed using grey correlation.".

19) Table 2: What are the units of measurement? (e.g., Age at childbirth is easily seen as YEARS. But, what unit of measurement is "women's education" at 71.18?

Thanks to the reviewer for your valuable comments.

Based on your comments, to avoid misinterpretation, we have inserted a column in Table 3 to indicate the units of measurement of the variable.

20) Line 581: Fix legend (**p<.05  rather than **        p<.05)

Thanks to the reviewer for your valuable comments.

Based on your comments, we change the **   p< 0.5 in the sentence to ** p< 0.5. see line 580.

21) Table 9: Identify what columns C1, C2, and C3 are.

Thanks to the reviewer for your valuable comments.

C1, C2, and C3 are equivalent to group1, group2, and group3, respectively. As you suggested, to avoid ambiguity, we changed group1, group2, group3 to C1, C2, and C3.

C1: Higher burden of child support. This grouping states that "fertility rate = cash benefits * services and in-kind support * child-rearing ratio "

C2: Low unemployment and high child support burden. This group states that "fertility rate = services and in-kind support * tax incentives * ~unemployment rate * child-rearing ratio ".

C3: Low unemployment rate and low burden of child support. This group states that "fertility rates = cash benefits*~services and in-kind support*tax incentives*~unemployment*~ child-rearing ratio".

22) Line 754: Rather than percentages, report numbers (e.g., rather than report "57% of countries," report "12 of 21 countries.") Do that for all 3.

Thanks to the reviewer for your valuable comments.

Based on your comments, we change the percentages to numbers. The sentences are as follows.

It was found that cash benefit expenditures made the largest contribution to fertility improvement in 12 of 21 countries, relevant services and in-kind expenditures made the largest contribution to fertility improvement in 6 of 21 countries and tax concessions made the largest contribution to fertility improvement in 3 of 21 countries (Table 6).

23) Line 782: Replace "the" with "its"

Thanks to the reviewer for your valuable comments.

Based on your comments, we change the word " the " in the sentence to " its ". Now this sentence is " when China faces its population problem.".

24) Line 787: Replace "better" with "greater"

Thanks to the reviewer for your valuable comments.

Based on your comments, we change the word " better " in the sentence to " greater ". Now this sentence is " China's early formulation and improvement of family welfare policies can have a greater effect in encouraging fertility.".

25) Line 797 on: This conclusion does not seem to be supported by findings directly from this study, especially the conclusion dealing with the quality of youth employment. If this is a conclusion or an IMPLICATION drawn from using the findings of OTHER research, cite that research.

Thanks to the reviewer for your valuable comments.

We have amended this paragraph in response to your comments. The revised paragraph is as follows.

According to the macro-environment, dynamically adjust the amount of the three types of public expenditure on family welfare. From the results of Table 10, it can be seen that the mix of policies to promote fertility varies with unemployment and child support burdens. Under the background of low unemployment rate, we need to choose the type of combination of family welfare public expenditure according to the situation of child support burden. High child support burdens require increased spending on services and in-kind and tax incentives to increase fertility, while low child support burdens require increased spending on cash benefits and tax incentives. Different social contexts therefore require a constant refocusing of policy spending to respond effectively to the decline in fertility.
